# A Novel Technique to Improve Anastomotic Perfusion Prior to Esophageal Surgery: Hybrid Ischemic Preconditioning of the Stomach. Preclinical Efficacy Proof in a Porcine Survival Model

**DOI:** 10.3390/cancers12102977

**Published:** 2020-10-14

**Authors:** Manuel Barberio, Eric Felli, Raoul Pop, Margherita Pizzicannella, Bernard Geny, Veronique Lindner, Andrea Baiocchini, Boris Jansen-Winkeln, Yusef Moulla, Vincent Agnus, Jacques Marescaux, Ines Gockel, Michele Diana

**Affiliations:** 1IHU-Strasbourg, Institute of Image-Guided Surgery, 67000 Strasbourg, France; eric.felli@ihu-strasbourg.eu (E.F.); pop.raoul@gmail.com (R.P.); margherita.pizzicannella@ihu-strasbourg.eu (M.P.); vincent.agnus@ihu-strasbourg.eu (V.A.); 2Department of Visceral, Transplant, Thoracic and Vascular Surgery, University Hospital of Leipzig, 4107 Leipzig, Germany; boris.jansen-winkeln@medizin.uni-leipzig.de (B.J.-W.); yusef.moulla@medizin.uni-leipzig.de (Y.M.); ines.gockel@medizin.uni-leipzig.de (I.G.); 3Physiology Institute, EA 3072, University of Strasbourg, 67000 Strasbourg, France; bernard.geny@chru-strasbourg.fr; 4Department of Pathology, University Hospital of Strasbourg, 67000 Strasbourg, France; veronique.lindner@chru-strasbourg.fr; 5Department of Surgical Pathology, San Camillo Hospital, 00152 Rome, Italy; baiocchiniandrea@gmail.com; 6Research Institute against Digestive Cancer (IRCAD), 67000 Strasbourg, France; jacques.marescaux@ircad.fr (J.M.); michele.diana@ircad.fr (M.D.)

**Keywords:** esophageal cancer, esophageal resection, Ivor-Lewis procedure, optical imaging, anastomotic leak, hyperspectral imaging, fluorescence imaging, confocal laser endomicroscopy, ischemic preconditioning

## Abstract

**Simple Summary:**

Esophagectomy has a high rate of anastomotic complications thought to be caused by poor perfusion of the gastric graft, which is used to restore the continuity of the gastrointestinal tract. Ischemic gastric preconditioning (IGP), performed by partially destroying preoperatively the gastric vessels either by means of interventional radiology or surgically, might improve the gastric conduit perfusion. Both approaches have downsides. The timing, extent and mechanism of IGP remain unclear. A novel hybrid IGP method combining the advantages of the endovascular and surgical approach was introduced in this study. IGP improves unequivocally the mucosal and serosal blood-flow at the gastric conduit fundus by triggering new vessels formation. The proposed timing and extent of IGP were efficacious and might be easily applied to humans. This novel minimally invasive IGP technique might reduce the anastomotic leak rate of patients undergoing esophagectomy, thus improving their overall oncological outcome.

**Abstract:**

Esophagectomy often presents anastomotic leaks (AL), due to tenuous perfusion of gastric conduit fundus (GCF). Hybrid (endovascular/surgical) ischemic gastric preconditioning (IGP), might improve GCF perfusion. Sixteen pigs undergoing IGP were randomized: (1) Max-IGP (*n* = 6): embolization of left gastric artery (LGA), right gastric artery (RGA), left gastroepiploic artery (LGEA), and laparoscopic division (LapD) of short gastric arteries (SGA); (2) Min-IGP (*n* = 5): LGA-embolization, SGA-LapD; (3) Sham (*n* = 5): angiography, laparoscopy. At day 21 gastric tubulation occurred and GCF perfusion was assessed as: (A) Serosal-tissue-oxygenation (StO_2_) by hyperspectral-imaging; (B) Serosal time-to-peak (TTP) by fluorescence-imaging; (C) Mucosal functional-capillary-density-area (FCD-A) index by confocal-laser-endomicroscopy. Local capillary lactates (LCL) were sampled. Neovascularization was assessed (histology/immunohistochemistry). Sham presented lower StO_2_ and FCD-A index (41 ± 10.6%; 0.03 ± 0.03 respectively) than min-IGP (66.2 ± 10.2%, *p*-value = 0.004; 0.22 ± 0.02, *p*-value < 0.0001 respectively) and max-IGP (63.8 ± 9.4%, *p*-value = 0.006; 0.2 ± 0.02, *p*-value < 0.0001 respectively). Sham had higher LCL (9.6 ± 4.8 mL/mol) than min-IGP (4 ± 3.1, *p*-value = 0.04) and max-IGP (3.4 ± 1.5, *p*-value = 0.02). For StO_2_, FCD-A, LCL, max- and min-IGP did not differ. Sham had higher TTP (24.4 ± 4.9 s) than max-IGP (10 ± 1.5 s, *p*-value = 0.0008) and min-IGP (14 ± 1.7 s, non-significant). Max- and min-IGP did not differ. Neovascularization was confirmed in both IGP groups. Hybrid IGP improves GCF perfusion, potentially reducing post-esophagectomy AL.

## 1. Introduction

Esophageal cancer is ranked seventh worldwide among all cancers, in terms of incidence (572,000 new cases) and ranked sixth for mortality rates (509,000 deaths) [1]. The treatment is multidisciplinary and surgery plays a central role in the therapeutic strategy [2]. Esophageal resection is a major surgical high-risk procedure with considerable complication rates [3], which may negatively impact the overall oncological outcome as well as the quality of life of patients [4,5]. In particular, the incidence of anastomotic leak (AL) is relevantly higher, when compared to other gastrointestinal anastomoses [6]. The etiology of AL is multifactorial. Some risk factors to develop AL are not modifiable and include preexisting relevant comorbidities [7] (e.g., hypertension, renal failure, congestive heart failure, smoking), and probably most importantly the presence of substantial atherosclerosis [8]. Appropriate blood supply at the gastric conduit fundus (GCF) is one of the key factors to promote anastomotic healing and to reduce the risk of AL [9]. The formation of the gastric conduit implies the tubulation of the stomach, the sudden disruption of a large part of its vascular supply, with the conduit fed by the gastroepiploic arcade only (and smaller branches of the right gastric artery, if preserved). This results in relative ischemia, especially at the apical section of the gastric graft, the former fundus, which will constitute the proximal portion of the esophagogastric anastomosis. 

Ischemic gastric preconditioning (IGP) has been previously proposed to improve the relative ischemia of the GCF by enhancing local microcirculation [10,11,12]. IGP consists in the partial devascularization of the stomach, using either an endovascular or a laparoscopic approach, several days or weeks before gastric pull-up and esophagogastric anastomosis completion. Although IGP has shown promising results in experimental settings and is a recognized safe procedure, it has failed to show a significant AL rate decrease to justify its implementation in the clinical routine [10,11,12,13,14]. This probably results from the large heterogeneity in terms of adopted IGP protocols and importantly from a too short delay between IGP and esophagectomy. 

As clinical intraoperative appraisal of the gastrointestinal tract blood flow is not accurate [15], several technologies have been explored to evaluate gastrointestinal perfusion during esophagectomy [9,16,17,18]. Fluorescence angiography (FA), a technique requiring a near-infrared (NIR) camera and the injection of a fluorophore, has become increasingly available and has shown promising results [9]. However, FA is currently lacking a standardized quantification method. Florescence-based enhanced reality (FLER), allows for precise quantitative fluorescence angiography in the digestive tract, and it has previously shown its accuracy during experimental gastrointestinal procedures [19,20]. Hyperspectral imaging (HSI) is another intraoperative optical imaging modality, which allows for contrast-free tissue oxygen quantification. HSI has proven accurate in assessing intestinal perfusion intraoperatively [21,22]. However, both FLER and HSI allow blood flow estimation exclusively at the serosal side. Confocal laser endomicroscopy (CLE) is a high-resolution microscopic imaging modality, which has been successfully used to quantify mucosal microcirculation within the digestive tract [23,24,25,26]. 

In this study we introduce hybrid IGP, by using a simultaneous endovascular/laparoscopic approach. The aim of this experimental comparative survival study in the porcine model is to assess the efficacy of hybrid IGP, by measuring the mucosal and serosal blood supply intraoperatively, using quantitative optical imaging technologies, supported by robust perfusion biomarkers.

## 2. Results

The main results are represented in Figure 1. During the surgical procedures at both T0 and T21 the vital parameters remained stable.

### 2.1. T0

#### FLER and LCL 

At T0, there was no statistically significant difference in terms of time-to-peak (TTP) (9.5 ± 3 s in max-IGP group; 8.8 ± 3.5 s in min-IGP and 12.1 ± 10.9 s in the sham group) and local capillary lactate (LCL) (2.8 ± 1.9 mmol/L in max-IGP group, 1.6 ± 1.3 mmol/L in group min-IGP and 1.2 ± 0.7 mmol/L in sham group) among the three groups (Figure 1A,B).

### 2.2. T21

All animals were clinically stable during the survival phase. They gained weight and showed no sepsis-related signs. The control digital subtraction angiography (DSA) of the celiac trunk confirmed the persistent occlusion of the embolized vessels in both IGP groups and the absence of noteworthy alterations in the sham groups (Figure 2F). 

At laparotomy, adhesions were present in one animal, and were caused by a moderate wound site abscess with intraperitoneal involvement without any related organ injury. The other animals did not show any noteworthy adhesions. However, another animal presented 2 covered small perforations of the posterior gastric wall. Both complications involved animals of the max-IGP group. 

#### 2.2.1. Serosal Quantitative Optical Imaging Measurements: HSI and FLER 

The tissue oxygen saturation (StO_2_) value measured using HSI was significantly lower in the sham group (41 ± 10.6%) than in the min-IGP group (66.2 ± 10.2%, *p*-value = 0.004) and the max-IGP group (63.8 ± 9.4%, *p*-value = 0.006). No difference was found between min-IGP and max-IGP. 

Conversely, TTP assessed by means of FLER was significantly higher in the sham group (24.4 ± 4.9 s) when compared to max-IGP (10 ± 1.5 s, *p*-value = 0.0008). TTP was also inferior in min-IGP (14 ± 1.7 s) when compared to the sham group, but this difference did not reach any statistical significance. No difference was encountered between the two IGP groups (Figure 1C,E).

#### 2.2.2. Mucosal Quantitative Optical Imaging Measurement: CLE 

The functional capillary density area (FCD-A) index in the sham group (0.03 ± 0.03) was significantly inferior to both min-IGP (0.22 ± 0.02, *p*-value < 0.0001) and max-IGP (0.2 ± 0.02, *p*-value < 0.0001). No difference was found between min IGP and max IGP (Figure 1F).

#### 2.2.3. LCL Sampling 

LCL in the sham group (9.6 ± 4.8 mL/mol) was significantly higher than in min-IGP (4 ± 3.1, *p*-value = 0.04) and max-IGP (3.4 ± 1.5, *p*-value = 0.02). Min-IGP and max-IGP did not differ statistically (Figure 1E).

#### 2.2.4. Histology: Capillary Density Score and Eosinophil Count Score 

Mean capillary density score in the sham group (0 ± 0) was significantly lower than in min-IGP (2 ± 0.9, *p*-value = 0.0004) and in max-IGP (2.4 ± 0.5, *p*-value = 0.0001). No difference was found between the two IGP groups. Similarly, the eosinophils count score was significantly lower in the sham group (0.2 ± 0.4) than in min-IGP (1.8 ± 0.98, *p*-value = 0.01) and in max -IGP (2.2 ± 0.83, *p*-value = 0.004). No difference was found between the two IGP groups (Figure 3).

#### 2.2.5. Immunohistochemistry Quantification 

The immunohistochemical staining using CD31 and caldesmone confirmed the increase of inflammation and of newly formed vessels of both IGP groups as compared to the sham group (Figure 4).

## 3. Discussion

AL has a multifactorial etiology. However, anastomotic site perfusion is a controllable risk factor of leakage [9,16,17,18]. It has been previously demonstrated in a number of experimental studies [27,28,29,30] that IGP improves GCF perfusion, provided that a minimum delay of 2 to 3 weeks is observed between IGP and gastric conduit formation. Similarly, the results of the present study unequivocally demonstrate that IGP, applied 3 weeks prior to gastric tubulation, improves mucosal and serosal perfusion at the GCF, assessed by means of quantitative optical imaging technologies. These findings are supported by LCL levels, which are a robust surrogate of perfusion in the porcine digestive tract [19,20,21,22,24,26,31]. Additionally, the presence of incremented vessels and inflammatory cells (eosinophils) in both IGP groups represents an indirect sign of neovascularization [32,33]. The choice to count eosinophiles has multiple reasons. In fact, these cells have been characterized as proangiogenic actors in the last two decades [33]. In particular, they take part of the innate cells group that contribute to the production of important mediators of the angiogenic/lymphangiogenic factors [34]. Additionally, eosinophils are easily visible under light microscope even at low resolution and low magnification, given their unique bilobate shape with a cytoplasm characterized by a large number of granules confering a specific reddish coloration to these cells in the hematoxylin and eosin staining. Thus, compared with other immune cells, eosinophils are the main visible actors playing a major role in angiogenesis. The histological findings of increased vessel density were confirmed by the immunohistochemical staining (overexpression of the reliable mature vessel indicators, such as CD31 and caldesmone within the specimens of both IGP groups [35]). Those findings, together with the increased eosinophils’ count, demonstrate new vessel formation over a 21 days period, triggered by the hybrid ischemic preconditioning. 

Despite promising results in experimental studies, when translated to humans, IGP has failed to exhibit a clear advantage in terms of AL reduction following esophageal resection [10,11,12,13,14]. Yet, the clinical trials using IGP are highly heterogeneous in terms of technique (angiographic embolization [36,37,38] or laparoscopic vessels ligation [16,18,39,40,41,42]) and number of vessels involved, and they are mostly of retrospective nature. Importantly, the delay between preconditioning and esophagectomy is largely variable in those studies. This heterogeneity could account for the apparent clinical inefficacy of ischemic preconditioning. However, the severity of the anastomotic complications in patients who receive IGP seems to be milder as compared to patients who undergo esophagectomy without preconditioning [10,11,12,13,14]. 

Authors using an endovascular approach reported better results in terms of increased GCF perfusion [36] or even AL reduction [38] as compared to the ones using laparoscopy. In fact, the largest patient series using laparoscopic IGP describe shorter time intervals than 2 weeks [40,43]. This results from the potential adhesion formation that laparoscopic vessel ligation or complete gastrolysis might induce [42,44]. In this respect, performing esophagectomy 4 to 5 days after laparoscopic IGP represents a convenient strategy in order to reduce the risk of finding firm adhesions intraoperatively, hence potentially increasing the esophagectomy operative time and difficulty degree. On the other hand, when performing endovascular IGP, it was possible to mostly maintain a two-week delay with esophagectomy. Additionally, it has been observed in a clinical series that a short delay between IGP and surgery (<15 days) might have deleterious effects on GCF blood supply, whereas a 2-week interval seemed to be protective against AL [42]. 

Interestingly, another group [16] found an increment in terms of mucosal oxygen saturation, measured using an endoluminal spectroscopic probe, in patients undergoing esophagectomy 4 to 5 days after laparoscopic IGP. Conversely, the same group [41] found no neoangiogenesis, assessed through VEGF (vascular-endothelial-growth-factor) increase, in the gastric fundus of patients who underwent laparoscopic IGP 4 to 5 days prior to esophagectomy. These findings suggest that there is a transitory compensating mechanism against the relative ischemia generated with IGP, involving a provisional redistribution of microcirculation, possibly via shunts opening, in its initial phases (4–5 days). This compensation eventually consolidates through the progressive formation of new capillaries, which are detectable after 2 to 3 weeks, as confirmed by our study. 

The radiological IGP approach is less invasive than laparoscopy and is suitable to be performed several weeks prior to esophagectomy. However, the SGAs, which, together with the LGA, participate consistently in the gastric fundus vascular supply [45], are technically difficult to embolize. In fact, most authors performing angiographic IGP describe a direct embolization of the splenic artery, potentially increasing spleen infarction risk [37,38,46]. As a result, we introduced a hybrid laparo-angiographic technique, in which IGP is achieved mainly angiographically, while the SGAs are divided laparoscopically, resulting in a minimization of the surgical manipulation. Indeed, the lack of adhesions at T21 laparotomy seems to support our hypothesis, which needs to be verified in humans. 

No substantial difference was encountered in terms of perfusion between max- and min-IGP groups. An explanation might be that the majority of the gastric fundus blood flow is supplied by the SGAs and the LGA in humans [45]. This might be true in pigs as well, as a result the role of the LGEA (additionally embolized in max IGP) in terms of fundic perfusion could be minimal. However, the small sample size of our study could limit the significance of the results, therefore further studies with larger numbers are required in order to precisely understand the required extent of IGP. Nevertheless, min-IGP showed a higher safety profile than max-IGP, suggesting a better suitability for a potential clinical translation. Hybrid IGP might be an option for patients presenting resectable esophageal cancer and it could be performed in the context of the staging phase, which usually takes place at least 3 weeks before esophagectomy. 

The merits of our study lie in the innovative approach, the survival design, and the robust methodology, implementing cutting-edge optical imaging technologies, allowing to quantify real-time perfusion.

The main downside is represented by limiting the surgical procedure to gastric conduit formation without esophagectomy and appraisal of the AL rate. This choice was taken considering that designing AL rate as primary endpoint, much larger groups would have been required to obtain a sufficiently powered study. Additionally, during preparation experiments preceding this work, we noticed a high intraoperative mortality in animals undergoing esophagectomy, possibly due to bradycardia triggered by extreme vasovagal sensitivity. This might justify the fact that, to our knowledge, no survival experiments have been published using a porcine esophagectomy model.

## 4. Materials and Methods 

### 4.1. Animals Characteristics and Experimental Set-up

Seventeen adult male pigs (Large White) were included in the present study, which is part of the ELIOS protocol (Endoscopic Luminescent Imaging for Oncology Surgery), jointly approved by the local Ethical Committee on Animal Experimentation (ICOMETH No. 38.2016.01.085) and the French Ministry of Superior Education and Research (MESR) (APAFIS#8721-2017013010316298-v2). All animals were managed according to French laws for animal use and care, and according to the directives of the European Community Council (2010/63/EU) and ARRIVE guidelines [47]. One animal died during anesthesia induction and was excluded from the study. The final number of animals was 16 (mean weight: 35.2 ± 10.7 kg). 

A preoperative 24-hour fasting with free access to water was part of the standardized protocol. Premedication was administered 10 min before surgery, using an intramuscular injection of ketamine (20 mg/kg) and azaperone (2 mg/kg) (Stresnil, Janssen-Cilag, Beerse, Belgium). Intravenous propofol (3 mg/kg) combined with rocuronium (0.8 mg/kg) was used for induction. Anesthesia was maintained with 2% isoflurane. A single prophylactic intramuscular dose of 150 mg of Cefalexine (Rilexine^®^, Virbac, Carros, France) was administered intraoperatively. 

The animals were randomly divided into 3 groups depending on the type of IGP that they underwent on day 0 (T0) (Table 1). 

Successively, all animals survived for 3 weeks with unrestricted access to food and water. Until postoperative day 3, analgesia was maintained with daily intramuscular injections of 50 mg of Flunixine (Finadyne^®^, Intervet, Beaucouzé, France). Gastric ulcer prophylaxis was provided with a daily oral administration of Esomeprazol 40 mg (Laboratoire Arrow, Lyon, France). The animals were assessed daily by a veterinarian for any sign of pain or sepsis.

On day 21 (T21), via laparotomy, a gastric conduit was fashioned, and at the end of the procedure, the animals were humanely sacrificed with an intravenous injection of Pentobarbital Sodium (40 mg/kg) (Exagon^®^, AXIENCE, Pantin, France), under 5% isoflurane anesthesia. T0 and T21 procedures are explained in detail in the following sections and Figure 5 shows a schematic representation of the experimental flow. 

During the surgical procedures at T0 and T21 a restrictive fluid resuscitation therapy with a maximum of 1 L Ringer lactate solution per procedure was performed and the following vital parameters were constantly monitored: peripheral oxygen saturation (StO_2_), heart rate, expiratory carbon dioxide.

#### 4.1.1. T0 Procedure

##### Ischemic Gastric Preconditioning

The experimental procedures were performed in a preclinical hybrid operative room equipped with a robotic C-arm (Artis Zeego, Siemens Healthcare, Erlangen, Germany), thus allowing to perform endovascular and surgical procedures in rapid succession. 

In strictly aseptic conditions, a 5 French sheath was introduced into the right femoral artery under ultrasound guidance. A 5 French Cobra-2 catheter (Terumo Europe NV, Leuven, Belgium) was placed at the origin of the celiac trunk and a selective DSA was performed using 10 mL of contrast medium at a rate of 4 mL/s (VISIPAQUE 270, GE Healthcare, UK) (Figure 2A–E). 

*Group max-IGP* (*n* = 6): The left gastroepiploic artery (LGEA) was selectively catheterized via the splenic artery, using a 2.8 French microcatheter (PROGREAT^®^, Terumo Europe NV, Belgium). This artery was occluded approximately 2 cm after its origin using large-volume 0.020-inch diameter coils (Ruby^®^ Coils, Penumbra Inc., Alameda, CA, USA). The same microcatheter was then used to successively catheterize the right gastric artery (RGA) and left gastric artery (LGA). Both arteries were embolized using 300–500 µm calibrated microspheres (EmboGold^®^, BioSphere Medical, Roissy-en-France, France), diluted in a mixture of 50% contrast medium and saline solution, until stasis of flow was achieved. 

*Group min-IGP* (*n* = 5): The LGA was selectively catheterized using the 2.8 French microcatheter. The artery was embolized using the same procedure as above. 

*Group sham* (*n* = 5): The 2.8 French microcatheter was used to sequentially catheterize the LGA, RGA, LGEA, and perform super-selective angiographic runs in each of these arteries without embolization. 

For both IGP groups, at the end of the procedure, a celiac trunk DSA run was performed, in order to confirm the efficacy of the embolization. The catheters were then withdrawn, and a four-port laparoscopy was performed. In groups max-IGP and min-IGP, all short gastric arteries (SGAs) were divided using a laparoscopic bipolar vessel sealer (LigaSure™ Advance™, Medtronic, Dublin, Ireland), whereas a diagnostic laparoscopy was performed in group sham. 

##### Fluorescence-Based Enhanced Reality (FLER)

In all three groups, the future anastomotic region on the gastric fundus was laparoscopically marked using the tip of a sterile surgical marker (Surgical Site Mini-Marker, Covidien, Walpole, MA, USA). Consecutively, by means of a laparoscopic Babcock grasper, the stomach was stabilized, and care was taken to center the gastric fundus within the surgical scene. Successively, an intravenous injection of 0.2 mg/kg of indocyanine green (ICG) (Infracyanine^®^, Serb, Paris, France) was administered, and a fluorescence angiography was performed, using a full HD 30-degree near-infrared (NIR) laparoscope (D-Light P, KARL STORZ, Tuttlingen, Germany). The fluorescent signal was quantified using the previously described ER-PERFUSION software (IRCAD, Strasbourg, France), which allows to compute perfusion as time-to-peak (TTP) of the signal evolution pixel-by-pixel [20,22,46]. TTP results from the velocity of the fluorescence signal until it reaches its maximal intensity peak within the first 40 s following fluorophore injection, and it is expressed in seconds. This information is visually transformed into a perfusion heat map, which is overlaid onto the real-time video creating the enhanced reality effect, hence allowing for an accurate spatial localization of the perfusion information. 

##### Local Capillary Lactate (LCL) Measurement 

Following FLER, LCL values at the marked fundic region were measured using a portable lactate analyzer (EDGE, Apex Bio, Hsinchu City, Taiwan). By incising the seromuscular layer with laparoscopic scissors, a blood drop was obtained and aspirated through a port, using a sterile 2 mL Falcon tube (BD Biosciences, Frankling Lake, NJ, USA) connected to a motorized pipette filler and then transferred to the lactate analyzer stripe sensor, as previously described [31]. The serosa defect was sutured using a 3/0 polypropylene (Prolene, Ethicon, Sommerville, NJ, USA) thread. This was left several centimeters long and it served as a landmark to identify the future anastomotic site on the gastric fundus during the T21 procedure. 

#### 4.1.2. T21 Procedure

##### Gastric Conduit Formation 

After 3 weeks of survival period, the T21 procedure began with a diagnostic angiography of the celiac trunk to qualitatively evaluate the redistribution of gastric vascularization elicited by the IGP. 

Successively, a median laparotomy was performed, gastric vessels were divided using the LigaSure™ Advance™ system (RGA, LGA, LGEA in all groups and the SGAs only in the sham group), the sole RGEA was maintained. Using a surgical stapler (ENDO GIA™ equipped with 45 cm Black Reloads, Medtronic, Dublin, Ireland), an approximately 4 cm wide gastric conduit was created by starting from the lesser curvature (Figure 6A–C).

##### Serosal Quantitative Optical Imaging Measurements: HSI and FLER

Immediately after gastric conduit formation, HSI and FLER were used in rapid succession to quantify GCF perfusion. The HSI system (TIVITA^®^, Diaspective Vision GmbH, Pepelow, Germany) was placed at a distance of 35 cm from the surgical scene, and using an auto-static arm, the same NIR camera used on T0 was installed next to it (Figure 6D). The GCF, marked with a long suture at T0, was centered in the scene and an HSI acquisition was performed (roughly 10 s in duration). Successively, after ICG injection, FLER was performed as described for T0 (acquisition approximately 40 s). The perfusion at the GCF was quantified using the metrics provided by both systems: StO_2_ for HSI and TTP for FLER. In order to precisely identify the GCF area with both imaging modalities, the pictures generated with the two imaging modalities were superimposed on top of one another. However, since the resolutions of the HSI images (640 × 480 pixels) and of the NIR laparoscope (1920 × 1080 pixels) differ, a custom-made transformation algorithm, based on the moving least square method [48] distorted the HSI image to fit the FLER image (Figure 7). This process was completed intraoperatively and required approximately 30 s. 

##### Mucosal Quantitative Optical Imaging Measurement: Confocal Laser Endomicroscopy (CLE) 

As for the next step, 2 mL of 10% fluorescein (Fluocyne^®^, Serb, France) was injected intravenously. A small opening on the gastric conduit’s stapler line was performed and a CLE probe (GastroFlex™ UHD, Mauna Kea Technologies, Paris, France) with a 55 to 65 μm confocal depth and a 1 μm resolution was inserted and manually directed in correspondence with the gastric fundus, to scan the gastric mucosa. Approximately 30 second-long videoclips were generated. The videos were analyzed postoperatively using the IC Viewer software (version 3.8.6) (Mauna Kea Technologies, Paris, France). The software allows for the automatic recognition of elongated shapes emitting fluorescent contrast, as vessels and the tool identifies diameters ranging from half to twice the diameter of interest. Finally, it computes the functional capillary density area (FCD-A) index, which results by multiplying the mean capillary diameter by the total vessel length, and by dividing the result by the total image area (Figure 8). 

##### LCL Sampling

A full-thickness biopsy was excised in correspondence with the GCF and the resulting blood was sampled using the abovementioned portable lactate analyzer to obtain LCL values. 

##### Histology Scores: Vessel Count and Eosinophils Count

Sections were fixed in 10% neutral-buffered formalin and processed for histologic examination including paraffin embedding, sectioning, and staining with hematoxylin and eosin. By observing the slides at 5 high-power fields, the eosinophils and the capillary vessels were assessed by two pathologists (VL and AB), who were blinded for the study groups, using the following semi-quantitative score:

<5 = 0—normal;

5–10 = 1—slight increase;

11–15 = 2—mild increase;

>15 = 3—high increase.

The scores were assigned independently, and in case of discrepancies, a consensus was reached.

##### Immunohistochemistry Analysis

Sections from selected paraffin blocks for each specimen were used for immunohistochemical analysis. A descriptive analysis on 3 slides representative of each group (min-IGP, max-IGP, and sham) was performed by both pathologists and a consensus was reached. Slides of 4 um-thick tissue sections were incubated at room temperature in an antigen retrieval process (EDTA citrate buffer, pH 8.3, CC1 buffer), revealed with ‘Ultra View’ Universal DAB Detection kit and counterstained with a Hematoxylin solution (Ventana Roche Systems). They were treated on automate VENTANA-Benchmark-XT with the following antibodies: CD31 (mouse monoclonal, JC70 clone, Cell Marque Systems; pretreatment: CC1 64 min; dilution: 1/25 during 32 min), and caldesmone (mouse monoclonal, h-cD clone, Dako Agilent Technologies, prediluted during 20 min, pretreatment: CC1 36 min).

### 4.2. Sample Size Calculation and Statistical Analysis

The level of local capillary lactates was used as primary outcome and the sample size was calculated based on previous works from our group, in which intestinal ischemia had been studied [21]. Using a non-inferiority design, at least 4 animals per group were required to have a 95% chance of detecting a difference in the primary outcome measure as significant at the 5% level. 

The statistics were performed using the Prism 8 software (Graph Pad, San Diego, CA, USA). Parametric or non-parametric one-way ANOVA with multiple comparison test was used as appropriate. A *p*-value < 0.05 was considered statistically significant. 

## 5. Conclusions

In conclusion, hybrid IGP improves mucosal and serosal blood flow of the future anastomotic region on the gastric conduit fundus. Our work suggests that the sole embolization of the LGA associated to the laparoscopic division of the SGAs (min IGP) would elicit an efficacious ischemic preconditioning effect improving gastric fundus perfusion. Those promising results need to be interpreted with caution and a clinical translation is necessary to demonstrate the validity of our protocol in humans undergoing esophageal resection. 

## Figures and Tables

**Figure 1 cancers-12-02977-f001:**
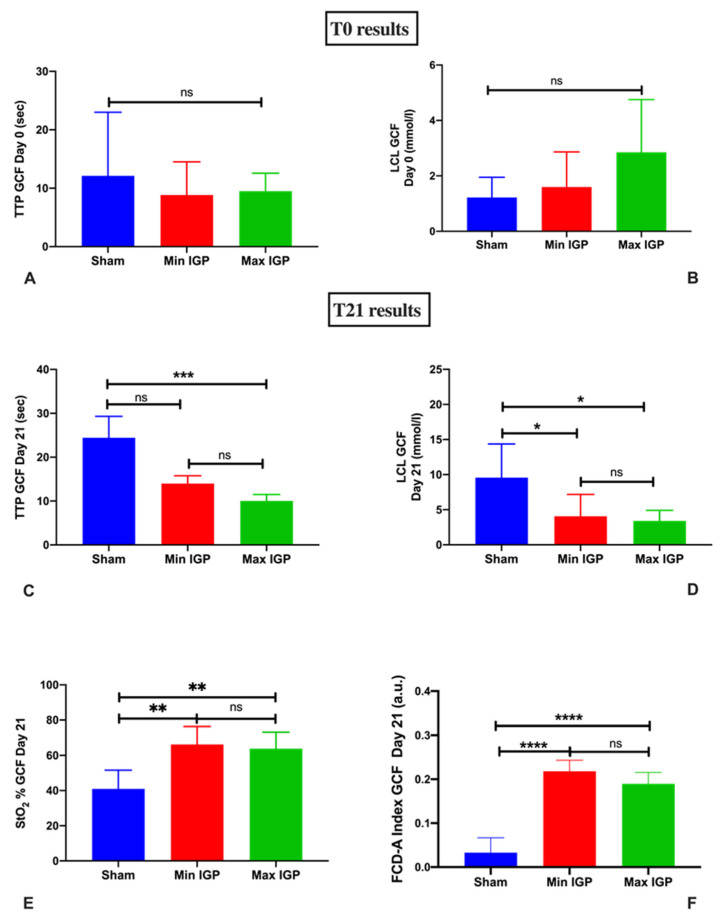
Graphical results overview: (**A**,**B**) show florescence-based enhanced reality (FLER) and local capillary lactates (LCL) levels at T0 immediately after completion of the ischemic gastric preconditioning (IGP). From (C–F), T21 results are displayed. All results point out the increase in terms of perfusion measured with FLER (**C**), hyperspectral imaging (HSI) (**E**), and confocal laser endomicroscopy (CLE) (**F**) and the reduction of LCL levels (**D**) of the IGP groups in comparison with the sham group. Although time-to-peak (TTP) (**C**) in the min IGP group is inferior compared to the sham group, this difference fails to reach any statistically significance, which is reached when comparing sham to max IGP. ns: *p*-value > 0.05; *: *p*-value ≤ 0.05; **: *p*-value ≤ 0.01; ***: *p*-value ≤ 0.001; ****: *p*-value ≤ 0.0001.

**Figure 2 cancers-12-02977-f002:**
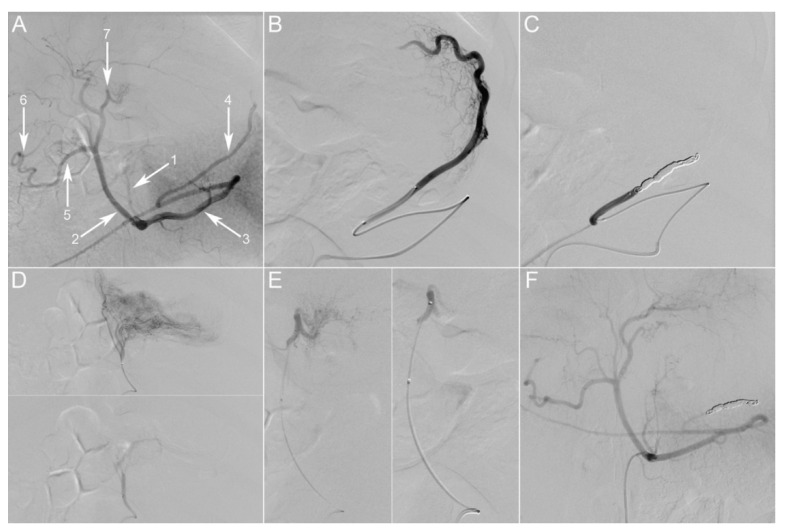
Gastric embolization. (**A**) Initial selective digital subtraction angiography (DSA) of the celiac trunk, posterior-anterior incidence. The main branches of the porcine celiac trunk are the following: left gastric artery (1), common hepatic artery (2), splenic artery (3). The common hepatic artery further divides into the gastroduodenal artery (5) and proper hepatic arteries. Gastric arterial supply consists of two anastomotic arcades: one on the lesser curvature between the left gastric artery (LGA) (1) and the right gastric artery (RGA) (7), the second on the greater curvature between the left gastroepiploic artery (LGEA) (4) and the right gastroepiploic artery (6). (**B**) Super-selective angiography through a microcatheter placed in the LGEA, depicting its vascular territory at the fundus. The artery was then occluded with coils (**C**). (**D**) Super-selective angiography through a microcatheter placed in the LGA, depicting its vascular territory on the lesser curvature and part of the fundus. The artery was then embolized using 300–500 µm particles. (**E**) Super-selective angiography through a microcatheter placed in the RGA, depicting its vascular territory on the lesser curvature and gastroesophageal junction. The artery was then embolized using 300–500 µm particles. (**F**) Selective DSA of the celiac trunk at 21 days post-embolization, showing persistent occlusion of embolized vessels. Picture taken at a factor 31 of magnification.

**Figure 3 cancers-12-02977-f003:**
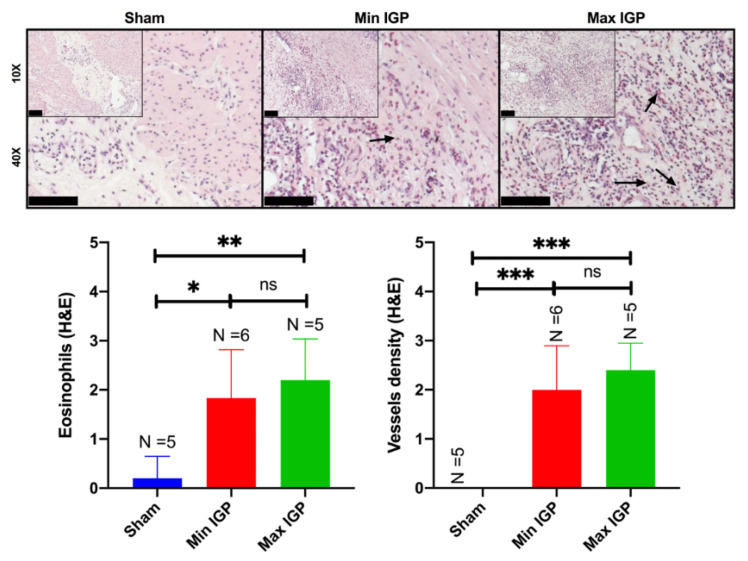
Histology In the (10× and 40×) images of the sham group, the normal gastric wall with the various layers and interstitial spaces of the muscle layer containing native capillaries is visualized. The reduction in blood supply leads to an increase in inflammatory cells in the lamina propria of the mucosa with hyperplasia of the foveola and congested vessels. As the reduction of arterial flow to the gastric wall at the level of the mucosa increases, also the edema increases with expansion of the lamina propria and relative reduction in the number of glands. The inflammatory infiltrate does not appear to increase, as confirmed in the semiquantitative score representing the eosinophils infiltration. The images of the min IGP group show, at muscular level, in the interstices between myocytes, the presence of edema and an extensive inflammatory lymphocytic and granulocytic infiltrate with numerous eosinophils and neoformed vessels which sometimes resemble granulation tissue (10× and 40×). This aspect seems to change in the max IGP images where the muscular inflammatory infiltrate seems to decrease while maintaining the increase of the newly formed vessels as if they were stabilizing. This results in an increase in vessels at the expense of a greater inflammatory infiltrate. Or rather: with a slight reduction in blood supply, we have a greater acute and chronic inflammation with eosinophils and neoformed vessels, while with a greater reduction in blood supply we have a reduction in acute and chronic inflammation with eosinophils but not in neoformed vessels. The arrows in the 40× histological images indicate the eosinophils showing their bilobulate shape and reddish staining. Scale bar 100 µm displayed at the left bottom of the 10× and 40× histopathological images. ns: *p*-value > 0.05; *: *p*-value ≤ 0.05; **: *p*-value ≤ 0.01; ***: *p*-value ≤ 0.001.

**Figure 4 cancers-12-02977-f004:**
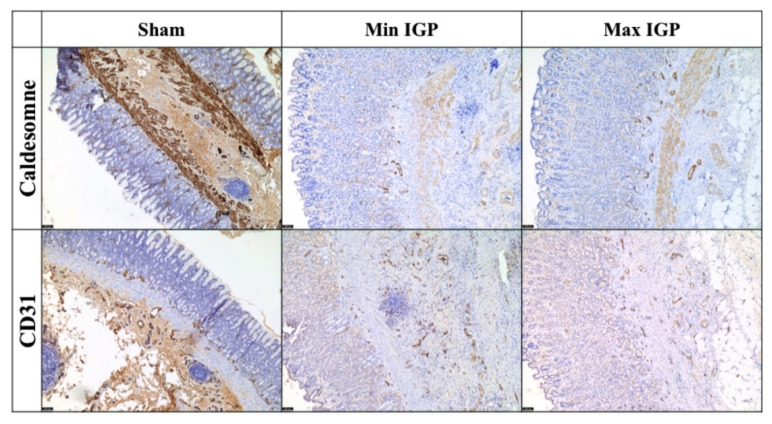
Immunohistochemical staining with CD31 (5×) shows an increase in neoformed blood vessels at the lower third of the mucosa with an edematous area near the muscularis mucosae. This area, which is not observed in the control sections, shows neoangiogenesis and inflammatory cells. Additionally, a dissolution of the lamina of the muscularis mucosae by edema and inflammation is visible. The immunostaining with caldesmone (5×) tends to overlap with CD31 and seems to more clearly define the edematous area with neoangiogenesis between the terminal portion of the gastric glands and the muscular bundles of the muscularis mucosae. The two immunocolorings are relatively overlapping and confirm the neoangiogenesis highlighted in H&E.

**Figure 5 cancers-12-02977-f005:**
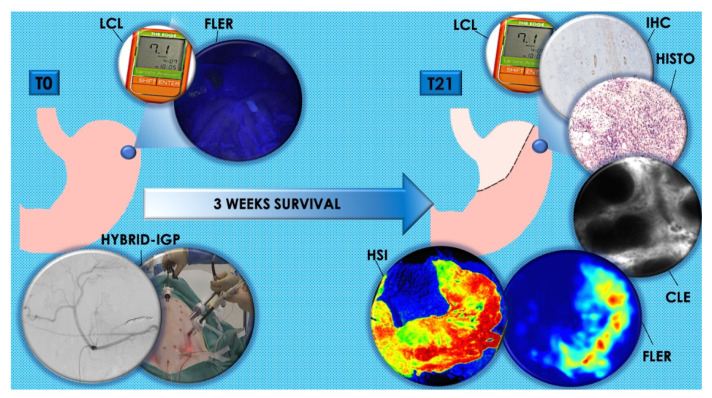
Description of experimental flow. At T0, hybrid IGP was performed in max-IGP and min-IGP groups, whereas the sham group underwent diagnostic angiography and laparoscopy. The blue spot on the stomach represent the future anastomotic site on the gastric fundus. All groups received a quantitative fluorescence angiography with FLER and LCL sampling at T0. At T21, an open surgical gastric conduit was formed and all groups underwent HSI, FLER, CLE, LCL sampling, histology, and IHC. Legends: T0: day 0; T21: day 21; hybrid IGP: hybrid ischemic gastric preconditioning; FLER: fluorescence-based enhanced reality; LCL: local capillary lactate; HSI: hyperspectral imaging; CLE: confocal laser endomicroscopy; Histo: histology; IHC: immunohistochemistry.

**Figure 6 cancers-12-02977-f006:**
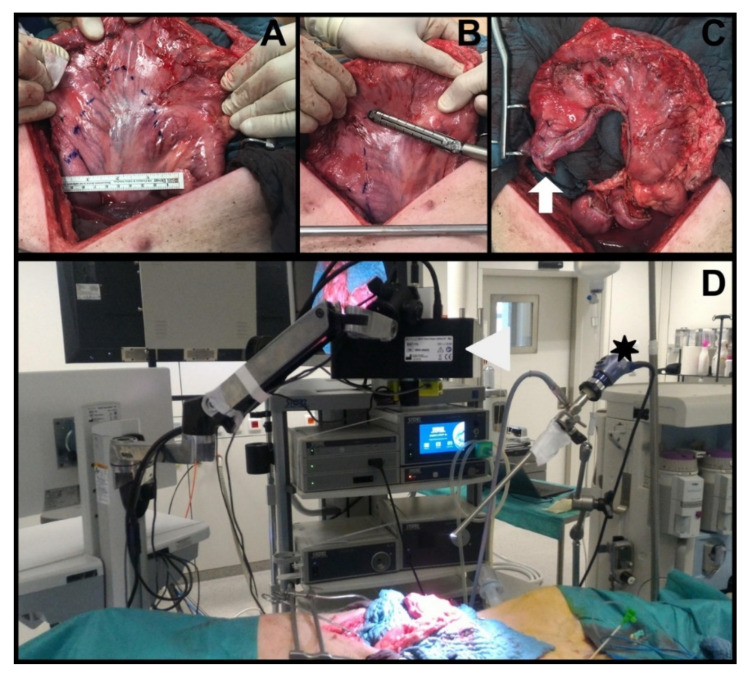
Experimental set-up: (**A**) T21 laparotomy with planning of the gastric conduit size; (**B**) gastric conduit formation using the Endo GIA™ linear stapler in an open fashion, starting at the lesser curvature; (**C**) completed gastric conduit with the future anastomotic site highlighted by the white arrow; (**D**) experimental set-up with both hyperspectral imager (white arrowhead) and near-infrared laparoscope (black asterisk) recording the surgical scene containing the gastric conduit.

**Figure 7 cancers-12-02977-f007:**
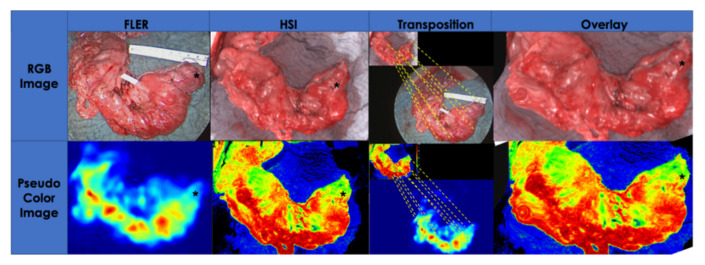
Superimposing HSI and FLER: The serosal perfusion quantification of the gastric conduit obtained using HSI and fluorescence angiography at T21 is compared by accurately superimposing the future anastomotic site on the fundus using the white light images generated by both cameras. Several identical features visible on both RGB images were manually selected by the operating surgeon until an algorithm, based on the least square principle, was able to overlay the images by distorting them in regions far away from the fundic region (black asterisk). After completion of this manual calibration, which took approximately 30 s, the quantification pseudo-color images of HSI and FLER were automatically superimposed by the algorithm, allowing for accurate co-localization and quantification of the gastric conduit fundus, using both metrics. Indeed, a degree of image distortion is visible in the left bottom corner of both overlay pictures.

**Figure 8 cancers-12-02977-f008:**
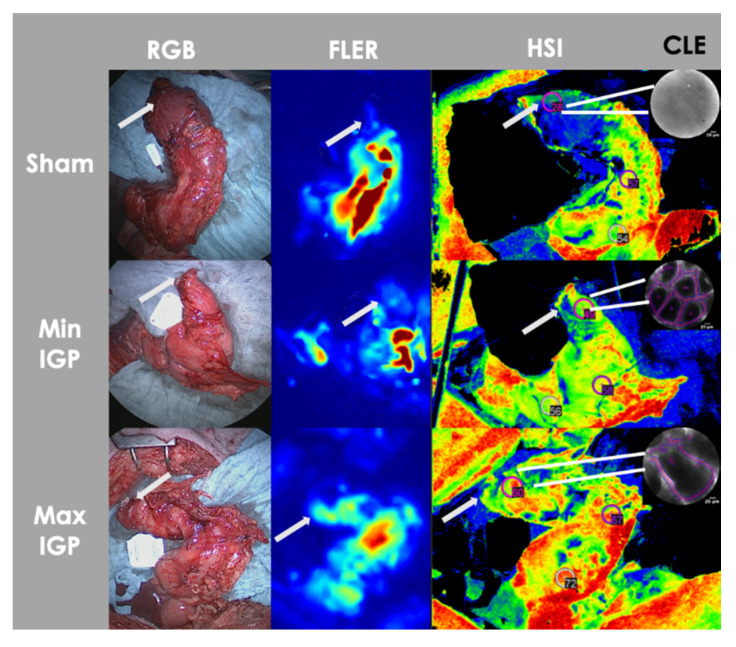
Serosal and mucosal optical imaging quantification. Visual representation of serosal (FLER, HSI) and mucosal (CLE) optical imaging quantification modalities of a gastric conduit of each group. The perfusion is visually incremented at the gastric conduit fundus/future anastomotic region (white arrow) in the IGP groups when compared to the sham group in all imaging modalities.

**Table 1 cancers-12-02977-t001:** Description of the study groups according to the intervention preformed on T0. T0: day 0; IGP: ischemic gastric preconditioning; RGA: right gastric artery; LGA: left gastric artery; LGEA: left gastroepiploic artery; SGAs: short gastric arteries.

Group	Hybrid IGP Intervention at T0
**Max IGP** (*n* = 5)	Angiographic embolization of RGA-LGA-LGEA Laparoscopic division of SGAs
**Min IGP** (*n* = 5)	Angiographic embolization of LGA Laparoscopic division of SGAs
**Sham** (*n* = 6)	Diagnostic angiography of RGA-LGA-LGEA Diagnostic laparoscopy

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
