# Peer review of "A Novel Technique to Improve Anastomotic Perfusion Prior to Esophageal Surgery: Hybrid Ischemic Preconditioning of the Stomach. Preclinical Efficacy Proof in a Porcine Survival Model"

_cancers, 2020, doi:10.3390/cancers12102977_

Round 1
Reviewer 1 Report
Dear Manuel,
it was a pleasure to go through your paper. Certainly, your group and you have put a lot of work into it.
I totally agree with your findings and congratulate you for this excellent work. Anastomotic healing in upper GI remains an unsolved problem and is responsible for major morbidity and mortality.
The scientific layout is good and you have applied great techniques.
As a clinician and upper GI surgeon, I would like you to address the following issues more in detail:
If I got it right, a esophagectomy was performed on day 21 hence why did you not choose to evaluate the anastomosis itself for all the tested parameters.
Also a evaluation of the anastomosis on day 1 3 5 and so on would have make sense e.g. using endoscopy.
Superimposed pictures of the future anastomotic site do not really represent the anastomosis itself
There is no difference between the min and max IGP group hence the conclusion should consider this
What are the clinical data in terms of leakage in all 3 groups?
Hope this is of some help.
Author Response
Dear Editors,
Dear Reviewers,
The authors are very grateful to the Reviewers for their encouraging feedback and their constructive observations aimed at improving the quality of our work. We hope you will find our answers/modifications to the text satisfactory and will agree to finally publish our work.
Please find enclosed a point-by-point answer to all the questions below.
Reviewer #1:
Dear Manuel,
it was a pleasure to go through your paper. Certainly, your group and you have put a lot of work into it.
I totally agree with your findings and congratulate you for this excellent work. Anastomotic healing in upper GI remains an unsolved problem and is responsible for major morbidity and mortality.
The scientific layout is good and you have applied great techniques.
A: Thank you for this positive feedback. We invested quite a lot of energies in this study and your comment make us proud. Certainly, such feedbacks encourage our group to continue future research in this field.
Q: As a clinician and upper GI surgeon, I would like you to address the following issues more in detail:
If I got it right, a esophagectomy was performed on day 21 hence why did you not choose to evaluate the anastomosis itself for all the tested parameters.
Also a evaluation of the anastomosis on day 1 3 5 and so on would have make sense e.g. using endoscopy.
A: Your observation is clinically absolutely relevant. Looking at the anastomotic leak rate would have been an ideal endpoint, increasing the clinical relevance of our protocol and enabling a smooth clinical translation. However, focus of the study was to evaluate the effects of the ischemic preconditioning on the gastric conduit. In fact, as you pointed out, at day 21 we performed the abdominal steps of the Ivory-Lewis esophagectomy, namely the gastric tube formation. We chose not to perform esophagectomy nor anastomosis. The reason for this choice is twofold. Firstly, if we would have chosen anastomotic leak as primary endpoint, given the reported rate in the modern series of 14-20%, we would have needed very large animal groups in order to have enough power in the statistical analysis (considering a superiority trial design, hypothesizing a success rate of 80% in the control and 90% in the preconditioning groups, with an alpha of 5% and a beta of 90%, 263 pigs per group would have been required). This would have been not feasible in terms of costs, since we decided to use the porcine model, given its similar size and vascular anatomy with humans, possibly ensuring an easier future translation of the experimental protocol. For this reason, we decided to investigate the perfusion, which is, thought to be one of the main causes of anastomotic leak, especially given the frequently faint blood-flow at the gastric tube’s tip. The second reason not to perform esophagectomy is that, during the preparation phase of this study, esophagectomies were performed in several pilot animals and we noticed that pigs are rather delicate once undergoing thoracotomy and esophagectomy, presenting a high rate of intraoperative cardiac arrests, possibly given by excessive sensitivity to inadvertant vagal stimulation. Our observation is proven by the fact that, to our knowledge, there are no published studies employing esophagectomy porcine models in the survival. Since aim of our study was to achieve a survival period of 21 days, in order to detect the effects of ischemic preconditioning on the gastric conduit blood-flow, we preferred to renounce performing esophagectomy, thus reducing the procedure related morbidity and mortality in pigs. We added an explanation concerning this in the paper, within the limitations, at the end of the discussion (pg. 8, line 271).
Q: Superimposed pictures of the future anastomotic site do not really represent the anastomosis itself
A: We thank you for this observation and we agree with it. In fact, since we did not perform anastomoses, we are not able to draw conclusions regarding the anastomosis itself. However, we investigated the very portion of the esophagogastrostomy often presenting a criticality in terms of perfusion, namely the future anastomotic site on the gastric conduit. Nevertheless, as you previously noticed, an endoscopic assessment is the best way to evaluate an anastomose, since you evaluate the anastomose at 360°. However, currently there is lack of reliable flexible endoscopes carrying the near-infrared imaging or hyperspectral imaging technology. For this reason, we had to use bulky experimental cameras for the serosal perfusion appraisal. We felt that in the context of our experimental setting, the superimposition of the perfusion information generated through HSI or fluorescence imaging, would have helped us to precisely identify the regions of interest for the capillary lactate and histological sampling.
Q: There is no difference between the min and max IGP group hence the conclusion should consider this
A: As appropriately pointed out from you, we emphasized the lack of perfusion difference between min und max IGP by adding a sentence in the discussion (pg. 8, line 258) and in the conclusion (pg. 14, line 461).
Q: What are the clinical data in terms of leakage in all 3 groups?
A: we are not able to answer this question, since as explained above our surgical intervention was limited to the creation of a gastric conduit.
Q: Hope this is of some help.
Personally, I found your observations rightful and absolutely relevant in view of a possible clinical translation of this protocol to humans. Thank you again.
Reviewer 2 Report
The manuscript is well written and easy to read. The issue is very attractive and incisive in clinical practice. The objective of the paper is clearly stated and the study design rigorously detailed and well done. The methods are adequately described.
However, the manuscript needs a few minor revisions.
The “Histology scores” section needs same clarifications. First: Why did the pathologists choose to count eosinophils only? Second: how is it possible to adequately evaluate the eosinophils count at 5 HPF in a inflammatory background? Could we have better detailed histological images representative of increased eosinophils number? Finally, could the Authors specify at line 205-207 what immunohistochemical staining has confirmed? (may be ‘increment of neoangiogenesis underlined by CD31 and Caldesmone immunostaining’? CD31 and Caldesmone are not ‘neoangiogenesis indicators’ overexpressed within the vessels but immunomarkers which underlined all vessels and, therefore, neo-vessels formed).
Could the Authors briefly explain why esophagectomy was not permormed?.
An interpretation about absence of statistical difference between the two IGP groups should be attempted even the limited number of cases may decrease the significance of the results, as already stated by the Authors.
We hope this study will help to decrease the anastomotic leaks in human pathology and we wish the Authors to continue these very important and interesting studies with rigorous competence.
Author Response
Rewiever #2
The manuscript is well written and easy to read. The issue is very attractive and incisive in clinical practice. The objective of the paper is clearly stated and the study design rigorously detailed and well done. The methods are adequately described.
However, the manuscript needs a few minor revisions.
A: Dear Reviewer #2, thank you for the encouraging feedback and the constructive observations.
Q: The “Histology scores” section needs same clarifications. First: Why did the pathologists choose to count eosinophils only?
A: We are grateful to the reviewer #2 for his/her important question. The eosinophils take part of the innate cells group that contribute to the production of important mediators of the angiogenic/lymphangiogenic factors(1). In particular, the eosinophils have been characterized as proangiogenic actors in the last two decades(2). Eosinophils’ infiltration is led by damage and inflammation which are stimulated by the preconditioning in our experiment. The hypoxia is promoted both from the blood hypoperfusion and the oxygen consumption by the eosinophils. Infiltrated eosinophils respond to the hypoxia increasing their proangiogenic potential acting as a main driver for the expression of CD300a(3). Additionally, eosinophils are easily visible at the microscope even at low resolution and low magnification. Thus, compared with other immune cells, eosinophils are the main visible actors that play a major role in the angiogenesis. Finally, their specific response for the hypoxia makes them an important target.
We have added some sentences and references 34 and 35 in the discussion session at pg. 7, line 207
Q: Second: how is it possible to adequately evaluate the eosinophils count at 5 HPF in a inflammatory background?
A: We would like to thank the reviewer #2 for his/her interesting question. In clinical setting, eosinophils are the major target for the angiogenesis due to their role and to the fact that are the most visible element in the Haematoxylin and Eosin staining even at the lowest magnification possible. This is true for instance in the Hodgkin’s lymphoma and especially true when an inflammatory background is present. Eosinophils present a unique bilobate shape with a cytoplasm that is characterized by a large number of granules that confer a specific staining to these cells. Eosinophils appear reddish in the Haematoxylin and Eosin staining. Their name, eosinophils is given by their affinity to the eosin. Due to these reason eosinophils don’t need a special staining and they can be clearly seen at low magnification. Finally, in an inflammatory context there is no recommendation in clinical setting for the eosinophils concerning the magnification except for the IgG4 disease for which the indication is 3 HPF and only for their manual counting(4). Thus, we consider that 5 HPF seems representative for the characterization of eosinophils infiltration with an inflammatory background.
Please, find some of the above-mentioned considerations in the discussion sessios (pg. 7, line 207)
Q: Could we have better detailed histological images representative of increased eosinophils number?
A: We agree with the reviewer that a better detailed histological image could highlight the increasing number of eosinophils. In fact, regardless the above consideration, a better zoom and a clearer panel figure should have been placed since the beginning. We apologize with the reviewer(s) and we thank to him/her for his important suggestion that let us improve the clarity of the histopathological results. Please find enclose in the reviewed version of this article a new figure 3, including both a 10X and 40X magnification. The figure text has been changed accordingly. The changes are highlighted in yellow.
Q: Finally, could the Authors specify at line 205-207 what immunohistochemical staining has confirmed? (may be ‘increment of neoangiogenesis underlined by CD31 and Caldesmone immunostaining’? CD31 and Caldesmone are not ‘neoangiogenesis indicators’ overexpressed within the vessels but immunomarkers which underlined all vessels and, therefore, neo-vessels formed).
A: We would like to thank the reviewer for her/his very interesting question. Caldesmone and CD31 confirmed the histopathological assessment of the increased vessel density. The reviewer is right affirming that these two markers are not specific for the angiogenetic process or molecular mediator, but they are good markers for the vessels in general. We would like to drive the attention that the study is based in the observation of the angiogenesis in a long-term period. We therefore aimed to observe the net increasing of only the mature vessels. We focused our attention on the increasing of the vessel’s density and not on the molecular mediators that could only potentially indicate the possible formation of new vessels such as VEGF. Indeed VEGF is the very first angiogenetic factor that is expressed after the decreasing of O2(5). This makes VEGF an important target for the study of the angiogenesis in an acute experimental setting. As pointed out, our study was based on 21 days survival. A marker such as VEGF would have overestimated or either underestimated the process in such a long-term study because is a target of both the new formed vessels and the overall signalling process that potentially induce new vessels. We therefore stained the vessels to furnish the evidence only of the well-structured and mature vessels for which Caldesmone and CD31 are good targets. Moreover, PECAM-1/CD31 and Caldesmone was shown to be involved in angiogenesis and the interactions of endothelial cell-cell adhesion molecules are important in the formation of new vessels(6, 7). Even though we consider VEGF a good target for the signalling process of the angiogenesis, evidences suggested that CD31 and Caldesmone were a better target given our experimental design.
Please, find some of the above-mentioned considerations in the discussion session (pg. 7, line 213).
Q: Could the Authors briefly explain why esophagectomy was not permormed?
A: Your question is absolutely rightful and has a great clinical relevance. As a demonstration of this also reviewer #1 asked the same question.
The main reason not to perform esophagectomy is that, during the preparation phase of this study, esophagectomies were performed in several pilot animals and we noticed that pigs are rather delicate once undergoing thoracotomy and esophagectomy, presenting a high rate of intraoperative cardiac arrests, possibly given by overreaction to iatrogenic vagal stimulation. Our observation is proven by the fact that, to our knowledge, there are no published studies employing the porcine esophagectomy model within a survival setting. However, aim of our study was to achieve a survival period of 21 days, in order to detect the effects of ischemic preconditioning on the gastric conduit blood-flow, as stated in the methods part under the smaple size calculation paragraph (pg. 14). As a result, we preferred to renounce performing esophagectomy, thus reducing the procedure related morbidity and mortality in pigs. Indeed, performing esophagectomies would have allowed us also to look at the anastomotic leak rate, resembling more the clinical situation. However, in order to design a sufficiently powered study, a large number of animals would have been required. We added an explanation of the choice not to perform esophagectomy in the paper, within the limitations, at the end of the discussion (pg. 8, line 271).
Q: An interpretation about absence of statistical difference between the two IGP groups should be attempted even the limited number of cases may decrease the significance of the results, as already stated by the Authors.
A: Thank you for this comment, we absolutely agree with it. The small sample size might have been a problem in order to understand the difference between the two IGP groups. We developed this part in the discussion paragraph, by adding two sentences marked in yellow (pg. 8, line 258).
We hope this study will help to decrease the anastomotic leaks in human pathology and we wish the Authors to continue these very important and interesting studies with rigorous competence.
Thank you again for your encouraging comments and constructive observation. We hope that our answers/ text modifications were satisfactory, and you will support the final publication of this work.
References
- Varricchi G, Loffredo S, Galdiero MR, Marone G, Cristinziano L, Granata F, et al. Innate effector cells in angiogenesis and lymphangiogenesis. Curr Opin Immunol. 2018;53:152-60.
- Nissim Ben Efraim AH, Levi-Schaffer F. Roles of eosinophils in the modulation of angiogenesis. Chem Immunol Allergy. 2014;99:138-54.
- Nissim Ben Efraim AH, Karra L, Ben-Zimra M, Levi-Schaffer F. The inhibitory receptor CD300a is up-regulated by hypoxia and GM-CSF in human peripheral blood eosinophils. Allergy. 2013;68(3):397-401.
- Deshpande V, Zen Y, Chan JK, Yi EE, Sato Y, Yoshino T, et al. Consensus statement on the pathology of IgG4-related disease. Mod Pathol. 2012;25(9):1181-92.
- Logsdon EA, Finley SD, Popel AS, Mac Gabhann F. A systems biology view of blood vessel growth and remodelling. J Cell Mol Med. 2014;18(8):1491-508.
- DeLisser HM, Christofidou-Solomidou M, Strieter RM, Burdick MD, Robinson CS, Wexler RS, et al. Involvement of endothelial PECAM-1/CD31 in angiogenesis. Am J Pathol. 1997;151(3):671-7.
- Zheng PP, Severijnen LA, van der Weiden M, Willemsen R, Kros JM. A crucial role of caldesmon in vascular development in vivo. Cardiovasc Res. 2009;81(2):362-9.
Round 2
Reviewer 1 Report
-